# A versatile genetic tool for post-translational control of gene expression in *Drosophila melanogaster*

**Sachin Sethi, Jing W Wang\***

Neurobiology Section, Division of Biological Sciences, University of California, San Diego, San Diego, United States

**Abstract** Several techniques have been developed to manipulate gene expression temporally in intact neural circuits. However, the applicability of current tools developed for in vivo studies in *Drosophila* is limited by their incompatibility with existing GAL4 lines and side effects on physiology and behavior. To circumvent these limitations, we adopted a strategy to reversibly regulate protein degradation with a small molecule by using a destabilizing domain (DD). We show that this system is effective across different tissues and developmental stages. We further show that this system can be used to control in vivo gene expression levels with low background, large dynamic range, and in a reversible manner without detectable side effects on the lifespan or behavior of the animal. Additionally, we engineered tools for chemically controlling gene expression (GAL80-DD) and recombination (FLP-DD). We demonstrate the applicability of this technology in manipulating neuronal activity and for high-efficiency sparse labeling of neuronal populations.

DOI: https://doi.org/10.7554/eLife.30327.001

## Introduction

Tools for precise spatial and temporal control of gene expression are essential for understanding how neuronal circuits develop and function. For example, limiting genetic manipulation of a target gene to a specific time and a defined neuronal population permits the separation of the developmental role of the gene from its contribution to circuit function in the adult stage. In the vinegar fly, *Drosophila melanogaster*, bipartite expression systems (GAL4/UAS, LexA/LexAop, QF/QUAS) provide a powerful means to control gene expression in a spatially selective manner (*Brand and Perrimon, 1993*; *Lai and Lee, 2006*; *Potter et al., 2010*). Several modifications of these expression systems have been made to permit temporal control over the exogenous transcription factors (GAL4, LexA or QF) (*Chan et al., 2015*; *McGuire et al., 2003*; *Osterwalder et al., 2001*; *Potter et al., 2010*). Much of the effort has been focused on using temperature or chemicals as means to control the gene expression systems. Temperature-dependent expression systems have been previously engineered by the direct fusion of a heat-inducible promoter to a gene of interest (*Lis et al., 1983*), or by using a temperature sensitive allele of GAL80, GAL80$^{ts}$. In the GAL4/UAS system, GAL80$^{ts}$ suppresses GAL4-induced gene expression at a low temperature (18°C) but not at a high temperature (29°C) (*McGuire et al., 2003*). Chemical-dependent tools include tetracycline-inducible systems (*Bello et al., 1998*; *Bieschke et al., 1998*), steroid hormone-inducible GAL4/LexA hormone receptor chimeras (*Han et al., 2000*; *Osterwalder et al., 2001*; *Roman et al., 2001*) and the quinic acid-inducible QS/QF/QUAS system (*Potter et al., 2010*). For example, an RU486-inducible GAL4 was made by fusing the GAL4 DNA-binding domain to the human progesterone receptor and the p65 transcriptional activation domain (*Roman et al., 2001*). However, there are limitations associated with the existing tools for temporal control of gene expression. First, temperature can

**\*For correspondence:**
jw800@ucsd.edu

**Competing interests:** The authors declare that no competing interests exist.

have a significant impact on the physiology and behavior of a fly (*Parisky et al., 2016*; *Sigrist et al., 2003*), which may prevent detection of the phenotype of interest. For example, temperature-dependent tools are unlikely to be suitable for the study of thermosensory circuits and related behaviors. Second, temporal control of gene expression using GAL80<sup>ts</sup> can only be achieved with transcription factors containing the GAD-activation domain, making it incompatible with a majority of the driver lines (LexA, QF and split-GAL4) that do not have the GAD domain. Third, application of the current chemical-dependent tools requires the generation of new transgenic stocks, such as new promoter-GAL4 lines. Additionally, RU486, a chemical used to induce gene expression in the Geneswitch system, has been reported to cause developmental lethality in flies with pan-neuronal expression of the RU486-sensitive GAL4 (*Li and Stavropoulos, 2016*).

We propose an alternative chemically inducible system, in which gene expression is controlled at the post-translational stage, making it compatible with the existing library of GAL4 stocks. We adopted the destabilizing domain (DD) derived from dihydrofolate reductase (ecDHFR) of *E. coli* to control protein stability in a ligand-inducible manner (*Cho et al., 2013*; *Iwamoto et al., 2010*), a strategy that has been used to control gene expression in mice and worms (*Cho et al., 2013*; *Iwamoto et al., 2010*; *Sando et al., 2013*). On fusing the destabilizing domain to a protein of interest, the chimeric protein is degraded by the proteasome, but its degradation is blocked by trimethoprim (TMP), a cell-permeable ligand for DD (*Figure 1A*) (*Iwamoto et al., 2010*). Thus, the protein of interest can be temporally controlled by TMP. A recent study demonstrated that TMP can regulate activity of the yeast I-SceI endonuclease in *Drosophila* larvae expressing the fusion protein I-SceI-DD (*Janssen et al., 2016*). Here, we characterized the efficiency and dynamics of this technology in vivo in the fly brain. We then used the DD technology to develop tools for mapping and manipulating neural circuits in *Drosophila*. As a proof of its utility, we fused DD to GAL80 and controlled GAL4-dependent gene expression in a TMP-dependent manner. We show that TMP can activate GAL80-DD to manipulate neuronal activity in behaviorally relevant sensory neurons. Additionally, by fusing DD to the FLP recombinase, we devised a strategy to control the recombination frequency within a neuronal population by controlling the concentration of TMP in fly food. We further used the destabilized FLP recombinase to refine the expression pattern arising from the intersection of two transgenic lines by temporally limiting the availability of TMP. In summary, we present a chemically inducible system optimized for neurogenetics in *Drosophila* with broader utility than comparable tools.

## Results

### Destabilized GFP

We first tested whether the ecDHFR-derived destabilizing domain (DD) can be used to control GFP expression levels. DD was genetically fused to the C-terminus of GFP and cloned into a 10XUAS construct (*Figure 1A*, *Figure 1—figure supplement 1*) to make *UAS-GFP-DD* transgenic flies. We reasoned that GFP expression should be conditioned on both the presence of the transcriptional activator GAL4 and the availability of the stabilizing ligand TMP. The expression of GAL4 in select neuronal populations affords spatial specificity. Feeding these flies with TMP at a specific time could provide a temporal control of GFP expression.

Using the pan-neuronal *nsyb-GAL4* to drive GFP-DD expression, we observed robust GFP expression throughout the brain of adult flies fed with TMP (*Figure 1B*). In the absence of TMP, GFP expression was low throughout the brain; this is consistent with the previous studies in mice and nematodes showing that unbound DD is an effective tag to mark the fusion protein for degradation (*Cho et al., 2013*; *Sando et al., 2013*) (*Figure 1B*). However, TMP levels may start to decline at the beginning of pupation, during which flies do not feed. To determine the efficiency of TMP-dependent GFP-DD stabilization during development, we measured GFP-DD expression throughout the brain from the larval to the adult stage (*Figure 1—figure supplement 2A*). We found substantial differences in GFP-DD expression between flies fed with solvent versus TMP during the larval (44 fold), 48 hr APF (19 fold) and three day old adult (27 fold) stages. Notably, even though flies do not feed for several days leading up to the late pupal stage (96 hr APF) and eclosion (<12 hr adult), we observed a five-fold difference in GFP-DD expression between flies raised on solvent versus TMP (*Figure 1—figure supplement 2A*). In summary, the TMP-inducible DD system is maximally effective

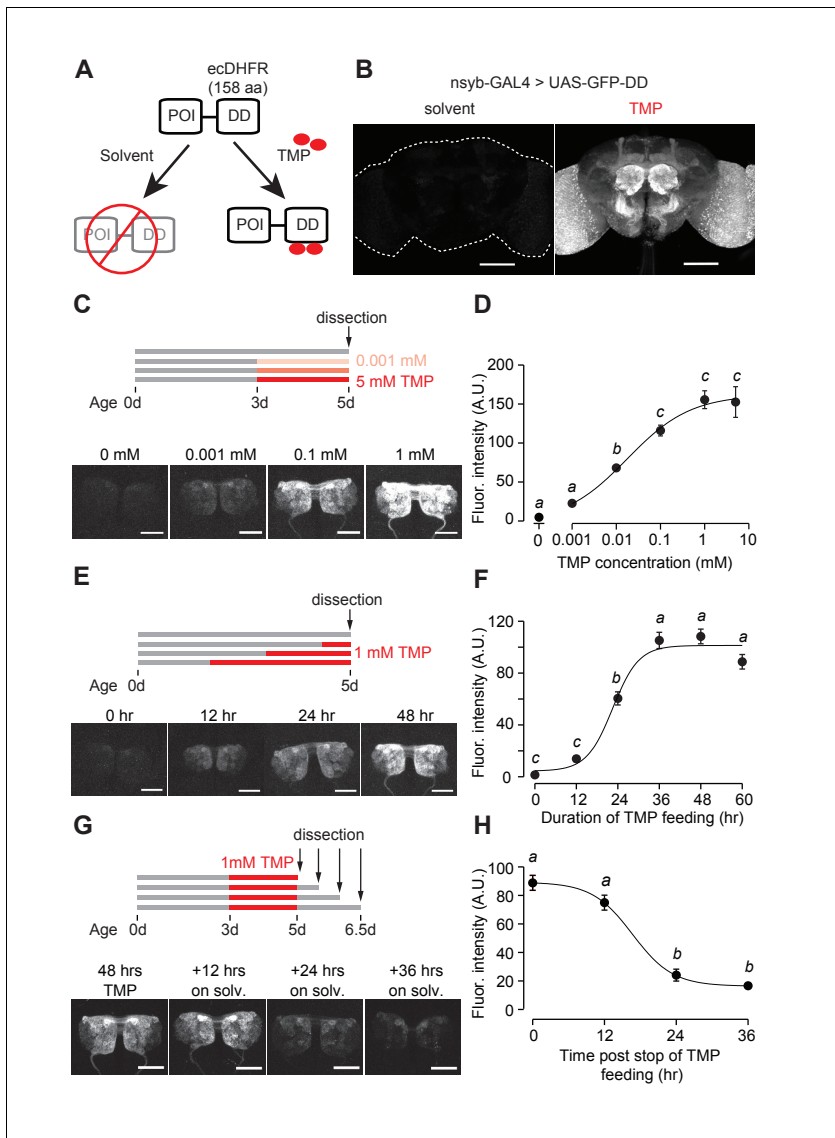

**Figure 1.** GFP-DD expression and degradation kinetics. (**A**) Schematic of the destabilizing domain (DD) system. ecDHFR = *E. coli*. dihydrofolate reductase, POI = protein of interest, TMP = trimethoprim. (**B**) TMP-dependent GFP expression in the adult brain. Flies were fed 1 mM TMP-containing food from embryo stage up to dissection. (**C, D**) Dose-dependent change in GFP-DD expression in the axonal terminals of olfactory sensory neurons. *Orco-Gal4, UAS-GFP-DD* flies were fed with TMP (0–5 mM) for 48 hr before dissection (n = 5–6, p<0.001, *F* = 41.37, one-way ANOVA with Tukey's post-hoc test). (**E, F**) GFP-DD expression is dependent on duration of TMP feeding. All flies were fed with fly food containing 1 mM TMP (n = 5–6, p<0.001, *F*= 87.34, one-way ANOVA with Tukey's post-hoc test). (**G, H**) GFP degradation kinetics. Flies were fed with 1 mM TMP for 48 hr and then switched to standard fly food. GFP-DD expression in the antennal lobe was observed at 12 hr intervals following the switch. (n = 8–10, p<0.001, *F* = 71.43, one-way ANOVA with Tukey's post-hoc test). Error bars indicate SEM. Significant differences between conditions (p<0.05) are denoted by different letters. Scale bar = 100 µm (**B**), 50 µm (**C, E, G**).
DOI: https://doi.org/10.7554/eLife.30327.002

The following source data and figure supplements are available for figure 1:

**Source data 1.** Dosage dependence and kinetics of GFP-DD expression in response to TMP feeding.
DOI: https://doi.org/10.7554/eLife.30327.010

**Figure supplement 1.** GFP-DD construct and sequence of the destablizing domain.
DOI: https://doi.org/10.7554/eLife.30327.003

**Figure supplement 2.** TMP-dependent GFP expression in the whole brain across developmental stages.
DOI: https://doi.org/10.7554/eLife.30327.004

*Figure 1 continued on next page*

*Figure 1 continued*

**Figure supplement 2—source data 1.** Effect of TMP feeding on GFP-DD fluorescent intensity across developmental stages.
DOI: https://doi.org/10.7554/eLife.30327.005
**Figure supplement 3.** TMP-dependent GFP expression in different tissues.
DOI: https://doi.org/10.7554/eLife.30327.006
**Figure supplement 3—source data 1.**
DOI: https://doi.org/10.7554/eLife.30327.007
**Figure supplement 4.** Effect of TMP on survival and behavior.
DOI: https://doi.org/10.7554/eLife.30327.008
**Figure supplement 4—source data 1.**
DOI: https://doi.org/10.7554/eLife.30327.009

during the larval, early pupal and adult stages. During the late pupal stage, however, the utility of the system may be limited by unavailability of TMP.

Because TMP-dependent protein stabilization acts through post-translational modification of protein levels, it should be compatible with any GAL4, LexA, QF or split-GAL4 line. To illustrate this principle, we visualized TMP-dependent GFP-DD expression across several different cell types in the adult brain using previously characterized driver lines (*Figure 1—figure supplement 3A*). We observed a significant difference in GFP-DD expression (ranging from 4 to 64 fold) between solvent and TMP feeding for every cell type that we tested. Maximum differences were observed in the axon terminals of sensory neurons (at least 45 fold). Interestingly, different cell types innervating the same anatomical region showed differential susceptibility to the DD system. For example, olfactory sensory neurons showed a much larger difference between TMP and solvent feeding (45 fold) compared to olfactory projection neurons (6 fold), even though both cell types innervate the antennal lobe. Similarly, PAM dopamine neurons showed a larger difference in GFP-DD expression (12 fold) compared to Kenyon cells (4 fold), even though both cell types innervate the mushroom body. Based on these results, we conclude that at least some of the differences between cell types arise from the variability in their proteasome activity levels. We also observed that the DD system was effective in non-neuronal tissues, such as ovaries (*Figure 1—figure supplement 3B*). Finally, to demonstrate that the DD system can be used in combination with other binary systems apart from GAL4/UAS, we generated a *13XLexAop-GFP-DD* transgenic fly line and observed similar TMP-dependent GFP expression in olfactory sensory neurons using the *Orco-LexA* driver line (*Figure 1—figure supplement 3D*).

We next carried out experiments to determine the kinetics and dynamic range of this chemical induction system using *Orco-GAL4* to drive GFP-DD expression in olfactory sensory neurons. Feeding adult flies with food containing varying concentrations (0–5 mM) of TMP for 48 hr resulted in a dose-dependent change in GFP expression in the antennal lobe (*Figure 1C,D*). The maximum GFP expression, induced by 1 mM TMP, was between 34 and 45 times higher than that of control flies fed with the solvent-containing food (*Figure 1D*, *Figure 1—figure supplement 3A*). Results from an experiment in which flies were fed for varying durations (0–60 hr) with food containing 1 mM TMP show that GFP levels increase initially but plateau within 36 hr (*Figure 1E,F*). To test if TMP-dependent GFP expression is reversible, we fed flies with food containing 1 mM TMP for 48 hr and then switched them to regular food (*Figure 1G*). We found that the GFP intensity was reduced by 73% within 24 hr (*Figure 1H*). In sum, using GFP as a test molecule, we show that genetic fusion of the ecDHFR-derived destabilizing domain confers instability to a protein of interest in *Drosophila*. Feeding flies with TMP can control protein levels in a reversible and dose-dependent manner with a large dynamic range.

We then investigated whether TMP has adverse effects on survival and behavior. Feeding adult flies with a defined medium containing TMP ranging from 0 to 10 mM did not have any detectable effect on their survival (*Figure 1—figure supplement 4A*). Moreover, feeding adult flies with 1 mM TMP for 48 hr did not alter their locomotion speed (*Figure 1—figure supplement 4C*) or their ability to locate an odor source (*Figure 1—figure supplement 4D*) during foraging. We then tested if TMP affects the development of the fly. Flies raised on TMP ranging from 0 to 10 mM throughout development were equally likely to survive to adulthood (*Figure 1—figure supplement 4B*). However, we

observed a delay in the developmental time for flies raised on 1 mM TMP (11.2 ± 0.1 days to eclosion) and 10 mM TMP (15.3 ± 0.3 days) as compared to solvent (10.2 ± 0.1 days) (*Figure 1—figure supplement 4B*). In summary, feeding adult flies TMP at 1 mM can induce GFP-DD expression at saturation level (*Figure 1D*) without any observable side effects on their survival or behavior. Notably, however, the same dosage of TMP delays development by about one day. Thus, we settled on the concentration of 1 mM TMP for further experiments except for cases when a lower level of induction was desirable.

## Destabilized GAL80

To evaluate the utility of the DD system for manipulating circuit function, we next investigated whether expression of GAL80 could be controlled by TMP. Binding of GAL80 to GAL4 prevents GAL4-mediated transcriptional activation in the GAL4/UAS expression system (*Lee and Luo, 1999*). We engineered a chemically inducible GAL80 by fusing DD to the C-terminus of GAL80. GAL80-DD was cloned downstream of a pan-neuronal promoter, *n-synaptobrevin* (*nsyb*). Addition of GAL80-DD to the GAL4/UAS expression system could allow TMP to control gene expression. Indeed, we found that *nsyb-GAL80-DD* was able to suppress GAL4-dependent GFP expression in olfactory sensory neurons (*Figure 2A*). This suppression of GFP expression in flies carrying the *Orco-GAL4*, *UAS-GFP* and *nsyb-GAL80-DD* transgenes was TMP-dependent (*Figure 2A*). This feature can be used to control gene expression to perturb neuronal function in a stage-dependent manner. For example, RNAi expression could be targeted to specific neurons during the adult stage by removing TMP from the food, which causes the degradation of GAL80. As a proof-of-concept experiment, we fed flies with TMP throughout development and up to 3 days post-eclosion (*Figure 2B,C*). When flies were moved from TMP-containing food to regular food, GFP expression started to increase after 24 hr, and peaked at 72 hr post-TMP removal (*Figure 2B1, C1*). In contrast, flies fed with TMP continuously, from embryo to adult, showed low GFP expression throughout the course of the experiment (*Figure 2B2, C2*). Furthermore, flies that were raised on regular fly food throughout showed high GFP expression (*Figure 2B3, C3*). To determine how soon after eclosion it is possible to activate gene expression, we raised flies on TMP during development up to eclosion and measured GFP expression in the brains of young adult flies (*Figure 2—figure supplement 1A*). We observed a difference between flies raised on TMP and those switched to solvent-containing food starting at one day post-eclosion (*Figure 2—figure supplement 1B,C*). However, we did not detect any GFP expression in flies younger than eight hours for either condition (*Figure 2—figure supplement 1B, C*). This lack of GFP expression even in the absence of TMP is most likely due to the time that is required to inactivate GAL80-DD upon TMP withdrawal (*Figure 1H*), although it is formally possible that the expression level of *Orco-Gal4* is low at eclosion. Either way, these results suggest that fusion of the ecDHFR-derived destabilizing domain to GAL80 permits TMP to control GAL80 activity, providing a chemically inducible system to control gene transcription in a temporal manner.

We further tested if GAL80-DD can be used to manipulate neuronal activity underlying behavior. We focused on the innate olfactory aversion to $CO_2$ in a T-maze assay. Olfactory aversion to $CO_2$ can be abolished by silencing Gr21a-expressing sensory neurons (*Suh et al., 2004*). We controlled the expression of tetanus toxin, a potent inhibitor of synaptic transmission (*Sweeney et al., 1995*), in Gr21a neurons using *nsyb-GAL80-DD* and TMP feeding (*Figure 2D*). *Gr21a-GAL4*-derived tetanus toxin expression was blocked in the presence of GAL80-DD when flies were fed TMP, but not when they were fed the solvent (*Figure 2D*). Accordingly, aversion to $CO_2$ was observed only when flies expressing *Gr21a-GAL4, UAS-TNT, nsyb-GAL80-DD* were fed with TMP, and not when they were fed with the solvent (*Figure 2E*). In comparison, control flies with GAL4 alone had high avoidance for both solvent and TMP feeding conditions. We observed a difference between solvent- and TMP-fed flies expressing *UAS-TNT, nsyb-GAL80-DD* (*Figure 2E*). However, the avoidance indices of both solvent- and TMP-fed *UAS-TNT, nsyb-GAL80-DD* flies were significantly higher than the avoidance index of *Gr21a-GAL4, UAS-TNT, nsyb-GAL80-DD* flies fed with the solvent (*Figure 2E*). In summary, we show that GAL80-DD can be used to manipulate GAL4-dependent expression of neuronal effectors and thereby alter the function of neuronal circuits underlying behavior.

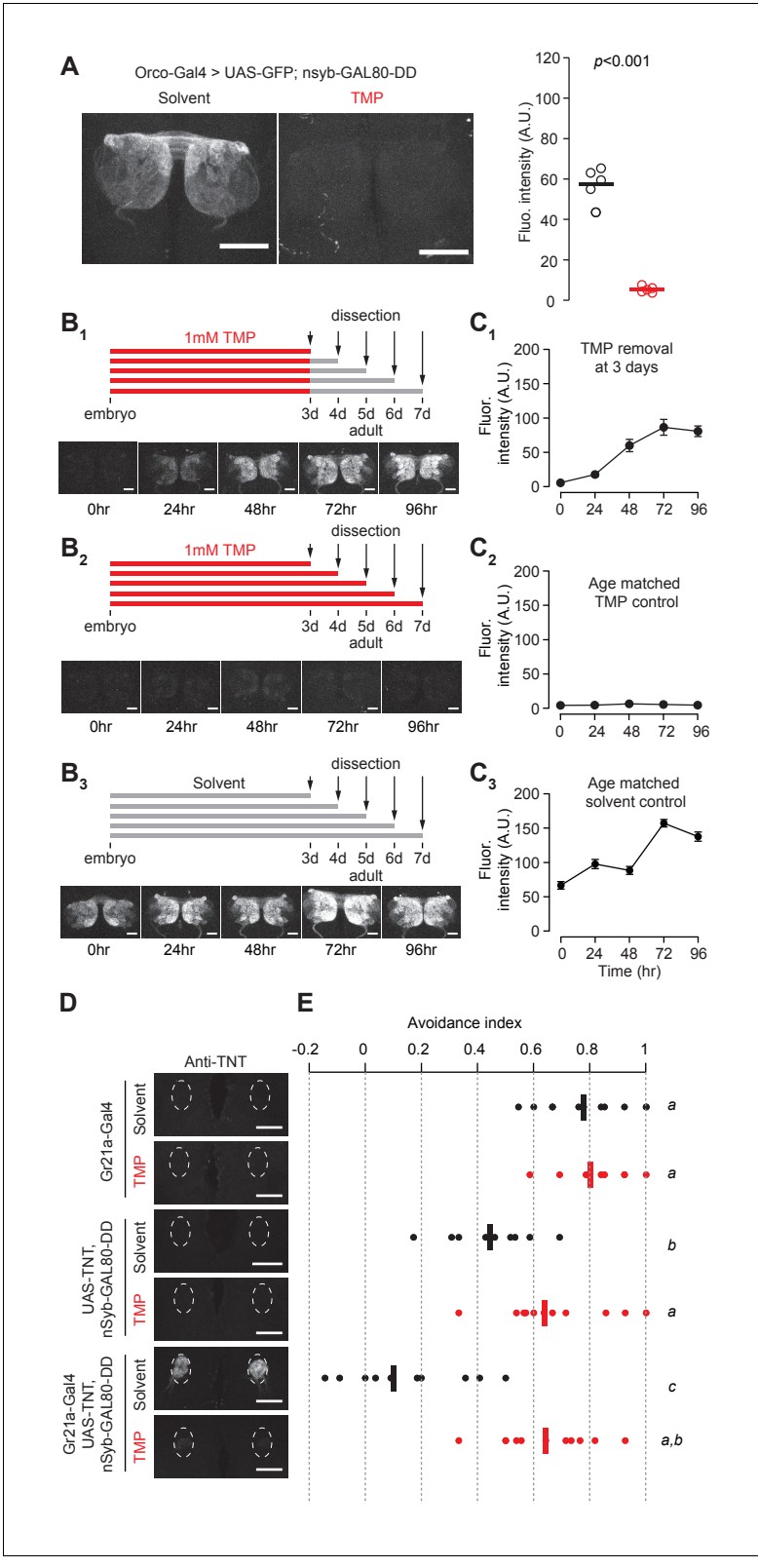

**Figure 2.** Chemically inducible control of GAL4-dependent expression by destabilized GAL80 (nsynaptobrevin-GAL80-DD). (**A**) GAL4-driven GFP expression in olfactory sensory neurons can be suppressed by destabilized GAL80 (*nsyb-GAL80-DD*) in a TMP-dependent manner (n = 5, unpaired t-test, two-tailed, *t* = 13.25). (**B, C**) GAL80-DD can be used to temporally control GFP expression. (**B₁, C₁**) *Orco-Gal4, UAS-GFP, nsyb-GAL80-DD* flies were

*Figure 2 continued on next page*

*Figure 2 continued*

fed with food containing 1 mM TMP up to 3 days post-eclosion, following which flies were switched to standard fly food and dissected for quantification. GFP expression was compared to flies fed with 1 mM TMP throughout ($B_2$, $C_2$) or solvent throughout ($B_3$, $C_3$) (n = 4–5). 0 hr time point in $C_1$ and $C_2$ represent the same sample. (D) Tetanus toxin expression in the V glomerulus of flies fed with 1 mM TMP or solvent. (E) $CO_2$ avoidance index of flies fed with 1 mM TMP or solvent. One arm of the T-maze contained 0.28% (v/v) $CO_2$ and the other arm had air. GAL80-DD can restore $CO_2$ aversion by suppressing TNT expression in the presence of TMP. n = 11 per condition, two-way ANOVA indicated a significant interaction between feeding condition and genotype, $F = 23.66$, $p < 0.001$. Significant differences between conditions ($p < 0.05$) are denoted by different letters (Tukey's post-hoc test). All flies were between 4–7 days old. Error bars indicate SEM. Scale bar = 50 µm (A), 25 µm (B,D).
DOI: https://doi.org/10.7554/eLife.30327.011

The following source data and figure supplements are available for figure 2:

**Source data 1.** Chemically inducible control of GAL4-dependent expression by GAL80-DD.
DOI: https://doi.org/10.7554/eLife.30327.014
**Figure supplement 1.** Functional characterization of GAL80-DD in early adults.
DOI: https://doi.org/10.7554/eLife.30327.012
**Figure supplement 1—source data 1.** Functional characterization of GAL80-DD in early adults.
DOI: https://doi.org/10.7554/eLife.30327.013

## Destabilized flippase

Flippase-mediated removal of a stop cassette has been widely used for lineage analysis and sparse neuronal labeling (*Lee and Luo, 1999*; *Marin et al., 2002*; *Wong et al., 2002*). Lineage analysis requires transient high-level expression of flippase (FLP) at specific developmental stages. On the other hand, sparse neuronal labeling requires low-level FLP expression in post-mitotic neurons for the stochastic removal of a stop cassette. Owing to the large dynamic range of the DD system, we reasoned it could be used to control FLP expression at different levels by varying TMP concentrations in fly food, thereby accommodating both sparse labeling and lineage mapping. The heat-shock promoter has been used previously to drive different levels of FLP expression by varying the duration of the heat-shock pulses. However, heat-shock driven FLP activity cannot be limited to a subset of cells due to the ubiquitous expression of the heat shock promoter. This limitation restricts the utility of *hs-FLP* for lineage analysis in an intersectional manner.

We fused DD to the C-terminus of FLP and incorporated the coding sequence into a 10XUAS construct (*10XUAS-FLP-DD*). We tested the destabilized flippase in olfactory projection neurons using *GH146-GAL4* to drive *UAS-FLP-DD* and a GFP stop-cassette reporter, *UAS(FRT.STOP) CD8GFP*. In these flies, stabilization of FLP-DD by TMP should permit FLP-mediated excision of the stop cassette, resulting in GFP expression in certain projection neurons. We observed that the number of GFP-positive olfactory projection neurons was correlated to the TMP dosage (*Figure 3A,B*). By varying the concentration of TMP (0.01–1 mM) in fly food, we could control the number of labeled projection neurons (*Figure 3A*). Furthermore, there were similar numbers of labeled neurons in both brain hemispheres for a given sample (*Figure 3B*). For flies fed with standard fly food without TMP, 42% of the brain hemispheres had only one GFP-positive cell (*Figure 3C*). This feature of FLP-DD can be used to generate single-cell clones at a reasonable probability for connectomics applications. As a proof-of-concept, we analyzed GFP-labeled neurons in the brains of 36 flies fed with solvent only. Out of the 72 brain hemispheres, 30 had only a single GFP-positive projection neuron (see *Figure 3D* for examples). In summary, dose-dependent expression of FLP-DD can be used to control the number of genetically manipulated cells within a population.

Restricting the activity of FLP-DD in a spatial and temporal manner should further refine expression patterns which arise from the intersection of two expression systems (eg. GAL4/UAS and QF/QUAS). To illustrate this principle, we focused on the intersection of *GH146-QF* and *NP21-GAL4*. It has been reported that the expression patterns for *NP21-GAL4* and *GH146-GAL4* overlap only in the DA1 lateral projection neurons (lPNs) in the adult brain (*Potter et al., 2010*), which we validated (*Figure 4A1,A2*). However, when UAS-FLP expression is driven by *NP21-GAL4*, the adult intersection pattern includes additional olfactory projection neurons, ellipsoid body neurons and neurons with cell bodies close to the lateral horn (*Figure 4B1,B2*). Similarly, when *QUAS-FLP* is driven by *GH146-QF*, the adult intersection pattern includes additional neurons (visual projection neurons in

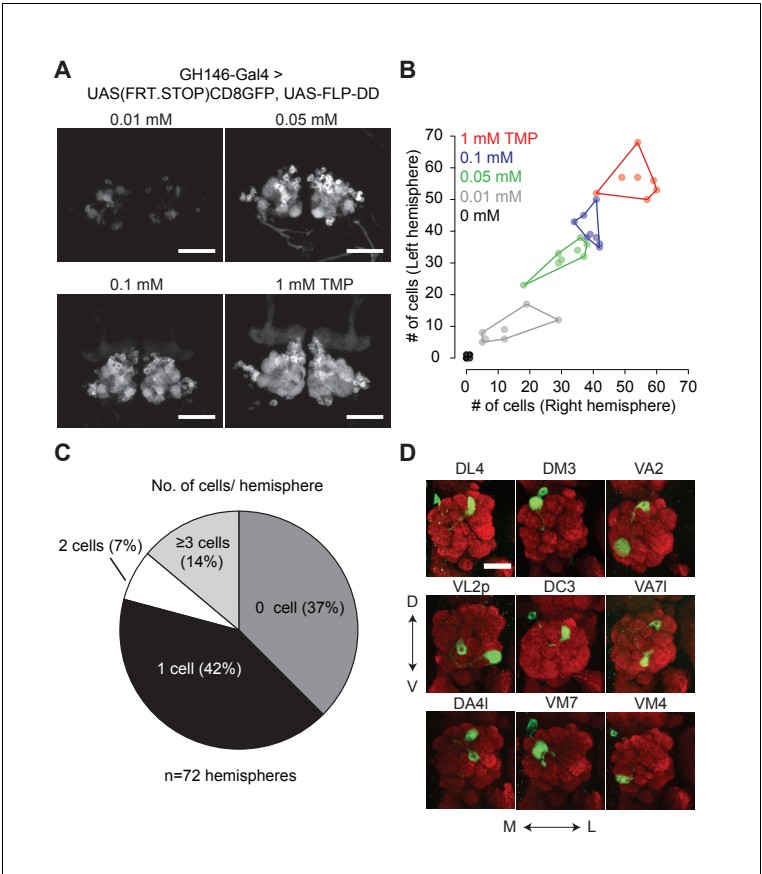

**Figure 3.** Chemical control of recombination frequency by destabilized flippase (*10XUAS-FLP-DD*). (**A**) GFP expression in a sub-population of olfactory projection neurons following excision of the STOP cassette by FLP-DD. Scale bar = 50 µm. (**B**) Number of GFP-positive projection neurons can be controlled by varying TMP dosage. The number of GFP-labeled cells within a sample is similar across both brain hemispheres. Each point represents number of cells in one brain. (**C**) Pie chart indicating the number of labeled projection neurons for flies fed with solvent. 42% of all hemispheres had a single GFP-labeled cell. (**D**) Examples of labeled single projection neurons (D - dorsal, V - ventral, M - medial, L - lateral). Red = anti Bruchpilot nc82, Green = GFP. Scale bar = 25 µm.
DOI: https://doi.org/10.7554/eLife.30327.015

The following source data is available for figure 3:

**Source data 1.** Chemical control of recombination frequency by FLP-DD.
DOI: https://doi.org/10.7554/eLife.30327.016

this case) (*Figure 4C1,C2*). This discrepancy between the overlap and the intersection patterns arises because of the broader expression patterns for *GH146-QF* and *NP21-GAL4* before the adult stage. Thus, the stop cassette is prematurely excised during development in neurons outside of the overlapping adult pattern. We reasoned that the adult expression pattern can be recapitulated by limiting TMP feeding to the adult stage thereby inactivating FLP during development. Indeed, when *UAS-FLP-DD* was expressed by the *NP21-GAL4* line and 1 mM TMP was fed to flies only during the adult stage, GFP expression was limited only to DA1 lPNs in the whole brain (*Figure 4D1,D2*). In comparison, flies fed with solvent alone did not have GFP expression in any neurons in the brain (*Figure 4E1,E2*). Furthermore, flies fed with 1 mM TMP throughout development have GFP expression in additional olfactory projection neurons (*Figure 4F1,F2*). We noted that the expression pattern in *UAS-FLP-DD* flies fed with 1 mM TMP throughout development was a subset of that observed with *UAS-FLP* flies (*Figure 4B,F*). It is possible that TMP levels decline in the fly brain during metamorphosis after the larvae stop feeding. In fact, similar results have been observed in the context of RU486-induced FLP activity (*Harris et al., 2015*). To mitigate this potential decline of TMP, we fed flies with 10 mM TMP throughout the larval stage and obtained a larger portion of the

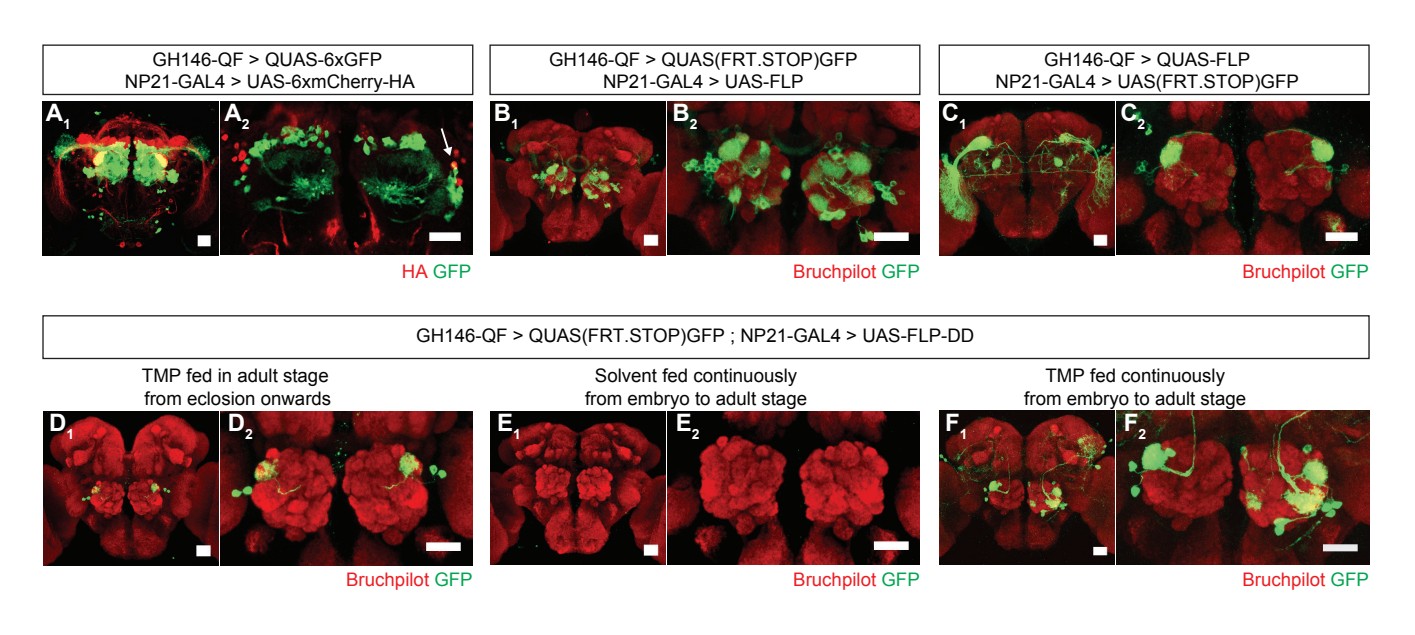

**Figure 4.** Refining intersection patterns by temporally limiting FLP-DD expression. (**A**) Z-stack projections showing expression patterns of *GH146-QF* (green) and *NP21-GAL4* (red). Both transgenic lines overlap in a single population of DA1 lPNs (arrow in A₂). Between one to three overlapping neurons can be observed across all samples. Antenna was ablated from the brain sample shown in A₂ to visualize projection neurons in the absence of sensory neuron axon terminals in the antennal lobe. (**B–C**) Intersection using constitutively expressed flippase generates expanded patterns with additional expression in other olfactory (**B**) or visual (**C**) projection neurons. (**D**) Temporally limiting FLP-DD expression by feeding 1 mM TMP exclusively in adult stage results in GFP expression only in DA1-lPNs. (**E**) No GFP expression is observed in the absence of TMP. (**F**) GFP expression in additional olfactory projection neurons can be observed using FLP-DD if TMP is fed continuously. Scale bar = 25 μm.

DOI: https://doi.org/10.7554/eLife.30327.017

The following figure supplement is available for figure 4:

**Figure supplement 1.** Intersection pattern between *GH146-QF* and *NP21-GAL4* in flies fed with 10 mM TMP continuously.

DOI: https://doi.org/10.7554/eLife.30327.018

*UAS-FLP* expression pattern (*Figure 4B*, *Figure 4—figure supplement 1*). In sum, we show that TMP can be used to limit FLP-DD activity temporally in a way such that the intersection pattern is identical to the overlap in the adult expression patterns.

## Discussion

Here we report a chemically inducible system to control gene expression in a dose-dependent and reversible manner. The DD system broadens the functionality of the *Drosophila* genetic toolkit as it provides an independent axis of control which can be used in combination with existing reagents. The DD system provides several advantages over existing chemical-dependent tools. First, TMP-induced DD stabilization is dose-dependent over several orders of magnitude of TMP concentration. This dose-dependency can be exploited for titration of in vivo gene expression levels. Here, we use this dose-dependent relationship to predictably alter the number of labeled projection neurons in the fly brain. Second, in contrast to existing chemical reagents, tools such as GAL80-DD can be combined with existing GAL4 lines to knockdown targeted genes by RNAi or to perform neuronal silencing screens in a temporally refined manner. Third, when TMP is withdrawn, the degradation kinetics of the DD fusion protein are most likely faster than those of the native protein. Thus, it is possible that the DD system offers fast temporal control in experiments which require reversible gene expression. Finally, it is worth noting that the cost of TMP is almost 150 times less than RU486 or quinic acid at their respective working concentrations, making the DD system conducive to large scale behavioral screens.

The applicability of the DD system for a given cell type is limited by two factors: (1) the cell type should have an active proteasome; (2) orally-fed TMP should be able to reach the cell. In the

absence of TMP, the level of proteasome activity in a given cell type may influence the residual level of a DD chimeric protein. For instance, low proteasome activity may result in high residual levels of the chimeric protein. It may be possible to reduce the residual expression by co-expressing components of the protein degradation machinery, similar to how co-expression of Dicer enhances RNAi efficiency (*Dietzl et al., 2007*). As it is, the background accumulation of the DD-fusion protein and the sensitivity of the downstream targets should be taken into consideration while designing experiments using the DD-system. As TMP is a cell-permeable ligand which can cross the blood-brain barrier, it should be accessible to all tissues during the adult and larval stages. However, the effectiveness of TMP in stabilizing the protein of interest during late pupal stages is reduced due to lack of feeding for several days leading up to this stage (*Figure 1—figure supplement 2*). This may lead to undesirable leaky expression when GAL80-DD is used with a GAL4 line that is highly expressed during the late pupal stage. Therefore, the level of gene expression controlled by GAL80-DD should always be determined using reporters before manipulating neuronal activity with effector transgenes. Due to the nature of chemical delivery, the utility of the DD-system is also limited to applications which can tolerate gene expression at a relatively slower rate. We anticipate that it will take roughly 24 hr to activate or inactivate the DD-system to an appreciable degree. However, the optimal delay from the start or stop of drug feeding is dependent on the level of gene expression required for a specific application. It may also be possible to achieve faster induction by using photocaged forms of trimethoprim (*Ballister et al., 2015*). During the course of our experiments, we observed a detrimental effect of the solvent DMSO (greater than 0.1%) on the survival of larvae (see Materials and methods). This toxicity can be circumvented by using a water-soluble form of TMP- trimethoprim lactate or by mixing pure TMP directly into the food. Finally, as trimethoprim is an antibiotic, experiments using the DD system should incorporate appropriate controls to rule out the effect of gut microbes on the phenotype of interest.

DD-based tools are conducive for mapping and manipulating neural circuits underlying behavior. We illustrate this concept by using destabilized GAL80-DD to chemically manipulate neural activity in olfactory sensory neurons. To our knowledge, GAL80-DD is the first construct that allows control of GAL80 activity in vivo by a small molecule. Conventional experimental designs utilize the heat-inducible GAL80<sup>ts</sup> to suppress GAL4-depdendent transcription at low temperatures. While GAL80<sup>ts</sup> may still be preferable in experiments that require low background expression, GAL80-DD is a useful alternative for experiments that are disrupted by temperature manipulations. To further illustrate the utility of the DD-system for circuit mapping, we engineered FLP-DD for sparse neuronal labeling at high-efficiency and temporally controlled genetic intersection. In a previous study, an RU486-inducible FLP recombinase was constructed by fusing it with the human progesterone receptor (Flp-Switch) (*Harris et al., 2015*). Although this construct can be chemically induced similar to FLP-DD, further experiments will be required to compare the efficacy and dose-dependency of the two recombinases. All DD-fusion proteins presented in this study are soluble molecules which function in the cytoplasm or the nucleus. However, previous studies have used ecDHFR-derived destabilized domains to conditionally alter membrane protein expression (*Iwamoto et al., 2010*). Therefore, we predict that this tool can be used to temporally control expression of membrane proteins such as ion channels and G-protein-coupled receptors.

In addition to chemically inducible forms of GAL80 and FLP, the DD technology can be used in flies for several other applications. DD can be knocked-in and fused to endogenous proteins to control their expression by limiting TMP feeding. This can be done using custom-designed genome editing strategies or by integration into the large number of available MiMIC sites within coding introns (*Venken et al., 2011*). DD can also be fused to a variety of effector genes for the purpose of inducible neuronal silencing or genome editing (*Maji et al., 2017*). Due to its inducible nature, GFP-DD can be coupled with knock-in GAL4 lines to compare gene expression in individual cells across time points spanning only a few hours, such as circadian fluctuation of gene expression. GFP-DD may also be useful as a sensor for proteasome activity.

In conclusion, we have developed a new set of tools for chemical control of gene expression in *Drosophila* which has broad-ranging applications and several advantages over existing tools of a similar nature. We characterized its efficiency and temporal limitations, and demonstrated its utility by engineering tools for chemical control of gene expression, recombination and neuronal activity.

# Materials and methods

**Key resources table**

| Reagent type | Designation | Source or reference | Identifiers | Additional information |
|---|---|---|---|---|
| genetic reagent (fly line) | nsyb-GAL4 | (*Riabinina et al., 2015*) | RRID:BDSC_51941 | |
| genetic reagent (fly line) | Orco-GAL4 | (*Kreher et al., 2005*) | RRID:BDSC_23292 | |
| genetic reagent (fly line) | UAS-GFP | | | |
| genetic reagent (fly line) | GH146-GAL4 | (*Stocker et al., 1997*) | RRID:BDSC_30026 | |
| genetic reagent (fly line) | Gr5a-GAL4 | (*Thorne et al., 2004*) | RRID:BDSC_57591 | |
| genetic reagent (fly line) | R58E02-GAL4 | (*Liu et al., 2012*) | RRID:BDSC_41347 | |
| genetic reagent (fly line) | P1a-split GAL4 | (*Hoopfer et al., 2015*) | | |
| genetic reagent (fly line) | MB434B-split GAL4 | (*Aso et al., 2014*) | RRID:BDSC_68325 | |
| genetic reagent (fly line) | Tdc2-GAL4 | (*Cole et al., 2005*) | RRID:BDSC_9313 | |
| genetic reagent (fly line) | MB247-GAL4 | (*Zars et al., 2000*) | | |
| genetic reagent (fly line) | UAS-(FRT.STOP)mCD8-GFP | (*Potter et al., 2010*) | RRID:BDSC_30032 | |
| genetic reagent (fly line) | UAS-(FRT.STOP)GFP.myr | | RRID:BDSC_55810 | |
| genetic reagent (fly line) | UAS-6XmCherry-HA | (*Shearin et al., 2014*) | RRID:BDSC_52267 | |
| genetic reagent (fly line) | QUAS-6xGFP | (*Shearin et al., 2014*) | RRID:BDSC_52264 | |
| genetic reagent (fly line) | 20XUAS-FLPD5 | (*Nern et al., 2011*) | RRID:BDSC_55805 | |
| genetic reagent (fly line) | GH146-QF | (*Potter et al., 2010*) | RRID:BDSC_30014 | |
| genetic reagent (fly line) | QUAS(FRT.STOP)GFP | (*Potter et al., 2010*) | RRID:BDSC_30134 | |
| genetic reagent (fly line) | NP21-GAL4 | (*Hayashi et al., 2002*) | RRID:BDSC_30027 | |
| genetic reagent (fly line) | Actin5C-GAL4 | | RRID:BDSC_4414 | |
| genetic reagent (fly line) | Orco-LexAVP16 | (*Lai et al., 2008*) | | |
| genetic reagent (fly line) | Gr21-GAL4 | (*Scott et al., 2001*) | | |
| genetic reagent (fly line) | UAS-TNT | (*Sweeney et al., 1995*) | | |
| genetic reagent (fly line) | 10XUAS-GFP-DD | this study | | see Materials and methods |
| genetic reagent (fly line) | 10XUAS-FLP-DD | this study | | see Materials and methods |
| genetic reagent (fly line) | nsyb-GAL80-DD | this study | | see Materials and methods |
| genetic reagent (fly line) | 13XLexAop-GFP-DD | this study | | see Materials and methods |
| antibody | Rabbit anti-GFP | Invitrogen | A-11122, RRID:AB_221569 | 1:200 |
| antibody | mouse anti-bruchpilot nc82 | DSHB | RRID:AB_2314866 | 1:50 |
| antibody | mouse anti-HA | Biolegend | 901501, RRID:AB_2565006 | 1:500 |
| antibody | rabbit anti-TeTx | Statens Serum Institut | POL 016 | 1:1000 |
| antibody | Alexa Fluor 488 anti-rabbit immunoglobulin G | Invitrogen | A-31628, AB_143165 | 1:100 |
| antibody | Alexa Fluor 647 anti-mouse immunoglobulin G | Invitrogen | A-21235, AB_2535804 | 1:100 |
| recombinant DNA reagent (plasmids) | pJFRC81 | *Pfeiffer et al., 2012* | Addgene_ 36432 | |
| recombinant DNA reagent (plasmids) | pJFRC95 | *Pfeiffer et al., 2012* | | |
| recombinant DNA reagent (plasmids) | nsyb-GAL4-hsp70 | *Riabinina et al., 2015* | Addgene_46107 | |
| recombinant DNA reagent (plasmids) | pCaSpeR-DEST5 | DGRC | 1031 | |
| recombinant DNA reagent (plasmids) | pAC-GAL80 | Addgene | 24346 | |

*Continued on next page*

*Continued*

| Reagent type | Designation | Source or reference | Identifiers | Additional information |
|---|---|---|---|---|
| recombinant DNA reagent (plasmids) | 10XUAS-GFP-DD | this study | | see Materials and methods |
| recombinant DNA reagent (plasmids) | 10XUAS-FLP-DD | this study | | see Materials and methods |
| recombinant DNA reagent (plasmids) | nsyb-GAL80-DD | this study | | see Materials and methods |
| recombinant DNA reagent (plasmids) | 13XLexAop-GFP-DD | this study | | see Materials and methods |
| chemical compound, drug | Trimethoprim | Teknova Inc., CA | T1205 | |
| software, algorithm | Igor Pro V6.0 | Wavemetrics, Inc. | RRID:SCR_000325 | |
| software, algorithm | Foraging assay behavior quantification | (*Zaninovich et al., 2013*) | | Code available from (*Zaninovich et al., 2013*) |
| other | Focusclear mounting reagent | Cedarlane Labs | FC-101 | |

## Fly husbandry

Flies were raised on standard fly food (unless otherwise noted) at 25°C in a 12:12 light-dark cycle. The following transgenes were used in this study - nsyb-GAL4 (*Riabinina et al., 2015*) (BDSC_51941), Orco-GAL4 (*Kreher et al., 2005*) (BDSC_23292), UAS-GFP, GH146-GAL4 (*Stocker et al., 1997*)(BDSC_30026), Gr5a-GAL4 (*Thorne et al., 2004*), R58E02-GAL4 (*Liu et al., 2012*), P1ª-split GAL4 (*Hoopfer et al., 2015*), MB434B-split GAL4 (*Aso et al., 2014*), Tdc2-GAL4 (*Cole et al., 2005*), MB247-GAL4 (*Zars et al., 2000*), UAS-(FRT.STOP)mCD8-GFP (*Potter et al., 2010*) (BDSC_30032) and UAS-(FRT.STOP)GFP.myr (BDSC_55810), UAS-6XmCherry-HA (*Shearin et al., 2014*) (BDSC_52267), QUAS-6xGFP(BDSC_52264)(*Shearin et al., 2014*), 20XUAS-FLPD5 (*Nern et al., 2011*)(BDSC_55805), GH146-QF(*Potter et al., 2010*) (BDSC_30014), QUAS(FRT.STOP)GFP (*Potter et al., 2010*) (BDSC_30134), NP21-GAL4 (*Hayashi et al., 2002*) (BDSC_30027), Actin5C-GAL4 (BDSC_4414), Orco-LexAVP16 (*Lai et al., 2008*), Gr21-GAL4 (*Scott et al., 2001*), UAS-TNT (*Sweeney et al., 1995*), 10XUAS-GFP-DD (this study), 10XUAS-FLP-DD (this study), nsyb-GAL80-DD (this study), 13XLexAop-GFP-DD (this study). See supplementary information for list of fly genotypes for every experiment.

## Transgenic fly generation

*Drosophila* codon optimized destabilized domain (DD) was synthesized with 5' *XhoI* and 3' *XbaI* overhangs by Genewiz, Inc. (La Jolla, CA). Plasmids were generated using standard protocols for PCR, restriction digestion and ligation.

## Destabilized GFP

To generate the 10XUAS-GFP-DD fly, DD was ligated to the c-terminus of GFP in the pJFRC81 vector (*Pfeiffer et al., 2012*). GFP was subcloned from the pJFRC81 plasmid using primer P1 and P2. DD was ligated to the c-terminus of GFP using the *XhoI* cut site. GFP-DD was ligated into the pJFC81 vector between the *PshAI* and *XbaI* cut sites. To generate the 13XLexAop2-GFP-DD fly, GFP-DD was cut from the 10XUAS-GFP-DD and ligated to the pJFRC95 plasmid (*Pfeiffer et al., 2012*) between the *NotI* and *XbaI* sites. Both GFP-DD constructs were transformed using phiC31 integrase mediated recombination into the attP2 landing site (*Groth et al., 2004*) by Genetic Services Inc. (Cambridge, MA).

## Destabilized GAL80

To generate the nsyb-GAL80-DD fly, DD was ligated to the c-terminus of GAL80. GAL80 was subcloned with 5' *EcoRI* and 3' *XhoI* overhangs from pAC-GAL80 plasmid (Addgene, Cambridge, MA. #24346) using primers P3 and P4. DD was subcloned from the 10XUAS-GFP-DD plasmid with 5' *XhoI* and 3' *AatII* overhangs using primers P5 and P6. GAL80-DD was triple ligated between *EcoRI* and *AatII* sites in the cut nsyb-GAL4-hsp70 plasmid (Addgene #46107) (*Riabinina et al., 2015*). The

resulting construct was transformed using phiC31 integrase mediated recombination into the VK00005 landing site (*Groth et al., 2004*) by Genetic Services Inc. (Cambridge, MA).

## Destabilized FLP

The 10XUAS-FLP-DD plasmid was generated by ligating DD to the c-terminus of FLPD5. FLPD5 was subcloned with 5' *NotI* and 3' *XhoI* overhangs from pCaSpeR-DEST5 (DGRC, Bloomington, IN. #1031) using primers P7 and P8. FLP was ligated between the *NotI* and *XhoI* sites in the cut 10XUAS-GFP-DD plasmid. The construct was transformed using phiC31 integrase-mediated recombination into the attP2 landing site by Bestgene Inc. (Chino Hills, CA).

## Primers

P1- GGAGTAGTCCCGATATTGGTTG
  P2- TTCATCTCGAGCTTGTAGAGCTCATCCATGCCGT
  P3- ATCATCGACAGCCGAATTCCAACATGGACTACAACAAGAGATCTTCG
  P4- GCGGCAATCAGGGAGATCTCGAGTAAACTATAATGCGAGATATT
  P5- CTGGTTTCCAAACTGATCGGTC
  P6- CGACGGTATCGATAGACGTCTATTAACGGCGCTCCAGAATCTCGAA
  P7- TACTTCAGGCGGCCGCGGCTGGAGGGTACCAACTTAAAAAAAAAAATCAAAATGCCACAATTTGATATATTATGT
  P8- ATCAGGGAGATCTCGAGTATGCGTCTATTTATGTAGGATG

## Recommended steps for generation of new POI-DD constructs

To generate new protein of interest (POI)-DD fusion constructs driven by UAS, the POI can be cloned in a non-directional manner (between *XhoI* restriction sites) or in a directional manner (between *NotI* and *XhoI*) (See *Figure 1—figure supplement 1*) using the UAS-GFP-DD plasmid as template. Note that for directional cloning between *NotI* and *XhoI*, the *XhoI* site upstream of Syn21 must be mutated during primer design (see P7 for example).The Syn21 sequence (21 bp) can be included within the primer if desired (see P7 for example).

## TMP feeding

Trimethoprim (Teknova Inc., CA) was maintained as a 100 mM stock solution in DMSO. To prepare food containing TMP for adult flies, standard fly food was heated to a liquid state. After cooling, TMP (or pure DMSO) was added to the food and vortexed to achieve a homogenous mixture of the required concentration. Food was poured into standard fly vials and allowed to solidify. Adult flies were transferred to new vials with TMP-containing food every 3 days. 1% DMSO was found to severely affect survival of larvae. Therefore, to feed flies with TMP from the embryo stage, pure TMP in powder form was mixed in fly food to attain the required concentrations. Detailed information on the feeding regimen for every experiment can be found in the supplementary information.

## Histology

Tissue samples were prepared for imaging using protocols that have been previously described (*Lin et al., 2013*). Tissues were dissected in cold PBS and fixed in 4% (w/v) paraformaldehyde for 3 min on ice in a microwave. Next, tissues were fixed in 4% (w/v) paraformaldehyde containing 0.25% Triton-X-100 for 3 min on ice in a microwave. Fixed tissues were placed in blocking solution (2% Triton X-100, 0.02% sodium azide and 10% normal goat serum in PBS) and degassed in a vacuum chamber for 6 × 15 mins to expel tracheal air. For the purpose of quantification in *Figures 1* and *2*, samples were not immunostained. All samples for a given experiment were prepared and imaged in parallel to allow for comparison among them. Rabbit anti-GFP (Invitrogen, Carlsbad, CA. A-11122, 1:200), mouse anti-bruchpilot nc82 (DSHB AB_2314866, 1:50), mouse anti-HA (Biolegend, San Diego, CA. 901501, 1:500) and rabbit anti-TeTx antibody (Statens Serum Institut, Denmark. POL 016, 1:1000) were used as primary antibodies in this study. Alexa Fluor 488 anti-rabbit immunoglobulin G (Invitrogen A-31628; 1:100) and Alexa Fluor 647 anti-mouse immunoglobulin G (Invitrogen A-21235, 1:100) were used as secondary antibodies. Brains were incubated in primary antibodies in dilution buffer (1% normal goat serum, 0.02% sodium azide and 0.25% Triton X-100 in PBS) for 48 hr at 4°C, rinsed for 3 × 15 mins in washing buffer (1% Triton X-100, 3% NaCl in PBS), incubated in

secondary antibodies in dilution buffer for 24 hr at 4°C, and rinsed again for $3 \times 15$ mins in washing buffer. Samples were mounted in Focusclear (Cedarlane Labs, Canada) between glass coverslips separated by spacer rings.

Samples were imaged with a 10X/0.3 or 20X/0.75 objective using a Zeiss LSM 510 confocal microscope to collect Z-stacks at 2 µm intervals. During the course of an experiment, the laser power and gain were held constant to allow for comparison among images from different experimental conditions. To quantify GFP expression, maximum intensity Z-projections were prepared using ImageJ (NIH). Average fluorescent intensity in the background was subtracted from the sample fluorescent intensity and the result was used as a proxy for GFP expression.

### T-maze assay

Flies were raised in standard fly food or food containing 1 mM TMP from embryo to adult stages up to the time of the experiment. Behavioral tests were performed as described previously (*Su et al., 2012*). About 30 flies were transferred from food vials into a 15 mL centrifuge tube (Fisher Scientific, Hampton, NH. 14959B) using a funnel. The tube containing the flies was connected to the T-maze apparatus and the flies were transferred into a horizontal elevator in the dark. Flies were held in the elevator for one minute before being pushed forward to choose between the test and the control arm. A fluorescent lamp was switched on at this point to phototactically draw flies out of the elevator. Flies were given one minute to choose between either arm, following which the elevator was retracted to separate the flies in the test arm from those in the control arm. The tubes serving as the test and the control arms were detached and flies in them were counted.

Flies were forced to choose between the control arm containing air and the test arm containing 0.28% $CO_2$. 400 µL of 10% $CO_2$ was injected into the test arm using a 10 mL syringe. The positions of the test arm and the control arm were alternated for each trial. The avoidance index was calculated as (no. of flies in the control arm - no. of flies in the test arm) /(no. of flies in the test arm + no. of flies in the control arm).

### Survival assay

Adult flies were raised on a defined medium (1 M sucrose, 1% agar) with 0–10 mM TMP from eclosion to death. Each experimental vial contained 5 males and 10 females. Flies were transferred to new vials every two days. Number of living flies was recorded every day. To quantify survival during development, 20 eggs were manually placed using forceps in a vial of fly food containing 0–10 mM TMP. Vials were observed daily to quantify the developmental timing for puparium formation and time to eclosion.

### Odor localization and locomotion assay

Odor localization ability and walking speed were measured using a setup described previously (*Root et al., 2011*; *Zaninovich et al., 2013*). Single flies were introduced in custom built chambers (60 mm diameter, 6 mm height) and tracked at 2 Hz under 660 nm LED illumination using custom software written in Labview (V.8.5, National Instruments, Austin, TX. Code available from *Zaninovich et al., 2013*). Wild type flies were fed with regular fly food containing 1 mM TMP or 1% DMSO for 48 hr before the experiment. The average walking speed of each fly during the first 50 s of each trial was determined using a custom macro with Igor Pro (V.6, Wavemetrics, Inc., Portland, OR). To perform the odor localization experiment, flies were transferred to starvation vials containing water with 1 mM TMP or 1% DMSO in Kimwipes 24 hr prior to the experiment. 1% apple cider vinegar in low melting agarose was used as the odor source. Latency to localization is defined as the elapsed time before a fly spends more than 5 s within 5 mm of the odor source.

### Statistical analysis

Statistical results (*p* value, effect size, n) are indicated in figure legends corresponding to each experiment. All statistical analyses were performed in Igor Pro (V.6, Wavemetrics, Inc.). Sample size for each experiment was pre-determined based on variation in experimental groups in pilot experiments. Most experiments were performed at least twice to confirm results. Data from one representative experiment is shown.

## Acknowledgements

We thank Dr. Chih-Ying Su and members of the Wang and Su labs for advice on experiments and comments on the manuscript; Dr. Steven Wasserman and Dr. John Belote for sharing fly lines; Dr. Barret Pfeiffer for sharing plasmids; Dr. Yishi Jin for sharing the confocal microscope facility.

## Additional information

### Funding

| Funder | Grant reference number | Author |
| --- | --- | --- |
| National Institute on Deafness and Other Communication Disorders | R01DC009597 | Jing W Wang |
| National Institute of Diabetes and Digestive and Kidney Diseases | R01DK092640 | Jing W Wang |
| National Institute of Mental Health | R21MH106958 | Jing W Wang |

The funders had no role in study design, data collection and interpretation, or the decision to submit the work for publication.

### Author contributions

Sachin Sethi, Conceptualization, Formal analysis, Investigation, Methodology, Writing—original draft, Writing—review and editing; Jing W Wang, Conceptualization, Supervision, Funding acquisition, Writing—review and editing

### Author ORCIDs

Jing W Wang  http://orcid.org/0000-0001-6291-5802

### Decision letter and Author response

Decision letter https://doi.org/10.7554/eLife.30327.021
Author response https://doi.org/10.7554/eLife.30327.022

## Additional files

### Supplementary files

• Supplementary file 1. Genotypes and feeding conditions listed by figure and experiment.
DOI: https://doi.org/10.7554/eLife.30327.019

• Transparent reporting form
DOI: https://doi.org/10.7554/eLife.30327.020

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
