## [Decision Letter]

Thank you for submitting your article "A versatile genetic tool for post-translational control of gene expression in *Drosophila melanogaster*" for consideration by *eLife*. Your article has been reviewed by two peer reviewers, and the evaluation has been overseen by Mani Ramaswami as Reviewing Editor and K VijayRaghavan as the Senior Editor. The following individuals involved in review of your submission have agreed to reveal their identity: Christopher J Potter (Reviewer #2).

The reviewers have discussed the reviews with one another and the Reviewing Editor has drafted this decision to help you prepare a revised submission.

Summary:

In the manuscript, Sethi et al. developed a clever strategy to reversibly regulate protein degradation. They introduce and characterize the use of destabilizing domains (DD) to 3 proteins of interest: GFP, GAL80, and FLPase. A C-terminal DD leads to proteasome degradation of the tagged protein, which can be prevented (in a dose dependent manner) by the presence of the antibiotic trimethoprim. The authors characterize the temporal dynamics of DD-mediated turnover, demonstrate it is amendable to a variety of neuronal tissues, and utilize it for behavioral analyses and intersectional expression studies. Particularly interesting is using UAS-FLP-DD instead of hs-FLP for the selective labelling of FLP-out constructs. to acutely manipulate neuronal activity and perform circuit analysis by adopting the destabilizing domain (DD) derived from ecDHFR of *E. coli*. The authors engineered GAL80-DD and FLP-DD, and tested their effectiveness using immunohistochemistry and behavioral assay. The results from proof-of-concept experiments are absolutely striking and clear. The manuscript is well written. The reagents presented here will be valuable additions to the *Drosophila* toolkit, and introduce a new method of temporal control over protein function.

Essential revisions:

1) The system may not work to regulate protein levels at time-scales < 24 hrs, or at pupal stages. This could limit some of its applications and, if so, then these should be formally detailed and prominently acknowledged.

2) Figure 2, Orco-GAL4, UAS-GFP, *nsyb*-GAL80-DD. TMP feeding was always performed from eclosion to 3 days. This allows ~2 days for adults to feed on TMP food. Is there GFP expression in newly eclosed flies? This should be checked. This is an important consideration, as it could suggest TMP+GAL80-DD may not function efficiently at late development stages (pupal), and the method needs a couple days post-eclosion of TMP feeding to stabilize protein levels in adult stages. This will influence the age at which adult animals are examined in many POI-DD experiments, as well as how future experiments are interpreted; for example, those that use the DD approach to ablate neurons.

---

## [Author Response]

Essential revisions:1) The system may not work to regulate protein levels at time-scales < 24 hrs, or at pupal stages. This could limit some of its applications and, if so, then these should be formally detailed and prominently acknowledged.

We thank the reviewers for bringing up this important issue and have added new experiments to characterize the efficacy of the DD system at pupal stages. We have revised the manuscript accordingly.

a) To address the temporal limitations of the system, we have added the following section to the Discussion:

“Due to the nature of chemical delivery, the utility of the DD system is also limited to applications which can tolerate gene expression on a relatively slower rate. We anticipate that it will take roughly 24 hours to activate or inactivate the DD-system to an appreciable degree. However, the optimal delay from the start or stop of drug feeding is dependent on the level of gene expression required for a specific application.”

b) To characterize the efficiency of the system during development, we measured GFP-DD expression throughout in the brain in the larval, early pupal (48 hr APF), late pupal (96 hr APF), at eclosion (<12 hr adult) and adult (3 day old) stages. We found substantial differences in GFP-DD expression between flies fed with food containing solvent versus TMP during the larval (44 times), 48 hr APF (19 times) and three day old adult (27 times) stages. Notably, even though flies do not feed for several days leading up to the late pupal stage (96 hr APF) and eclosion (<12 hr adult), we observed a five-fold difference in GFP-DD expression between flies raised on solvent versus TMP. This new data is now presented in Figure 1—figure supplement 2. We agree that it is important to inform readers of the smaller difference in the late pupal stage, which may not be enough to substantially manipulate a phenotype of interest. Accordingly, we have added the following section to the Discussion:

“However, the effectiveness of TMP in stabilizing the protein of interest during late pupal stages is reduced due to lack of feeding for several days leading up to this stage (Figure 1—figure supplement 2). This may lead to undesirable leaky expression when GAL80-DD is used with a GAL4 line that is highly expressed during the late pupal stage. Therefore, the level of gene expression controlled by GAL80-DD should always be determined using reporters before manipulating neuronal activity with effector transgenes.”

2) Figure 2, Orco-GAL4, UAS-GFP, nsyb-GAL80-DD. TMP feeding was always performed from eclosion to 3 days. This allows ~2 days for adults to feed on TMP food. Is there GFP expression in newly eclosed flies? This should be checked. This is an important consideration, as it could suggest TMP+GAL80-DD may not function efficiently at late development stages (pupal), and the method needs a couple days post-eclosion of TMP feeding to stabilize protein levels in adult stages. This will influence the age at which adult animals are examined in many POI-DD experiments, as well as how future experiments are interpreted; for example, those that use the DD approach to ablate neurons.

We agree with the reviewers that it is important to know whether GAL80-DD is functioning during the pupal stage. As mentioned above, we found that there is significant amount of TMP in late pupal stage to stabilize GFP-DD. Therefore, we expect that there may be sufficient TMP to power GAL80DD in newly eclosed *Orco-GAL4, UAS-GFP, nsyb-GAL80-DD* flies. We therefore repeated the experiment presented in Figure 2 by feeding the flies with TMP only up to eclosion and measured GFP expression at eclosion, 1 day and 2 days. Our new results show that there is a difference between flies raised on TMP and flies switched on to solvent containing food starting at one day post eclosion (Figure 2—figure supplement 1). However, we did not detect any GFP expression in flies younger than eight hours for either condition (Figure 2—figure supplement 1). This lack of GFP expression even in the absence of TMP is most likely due to the longer time (~24 hr) that is required to inactivate GAL80-DD upon TMP withdrawal, although it is formally possible that the expression level of Orco-Gal4 is low at eclosion. Nevertheless, potential leaky expression is an important issue that may affect data interpretation for future experiments. Therefore, we have added caution remarks in the Discussion (see above).